# OpenReview forum: "Reliability-Aware Preference Learning for LLM Reward Models"
_ICLR.cc/2025/Conference — ICLR 2025 Conference Withdrawn Submission_

### Official Review · Reviewer_Ks25 · 2024-11-02

**Soundness:** 2
**Presentation:** 1
**Contribution:** 2
**Rating:** 3
**Confidence:** 4

**Summary:**

This paper argues that a source of error in current reward models is that the collected preference judgments used to train them can reflect biases in annotator preferences during data collection. To verify this hypothesis, the authors collect a dataset (based on TruthfulQA), Length Incentivized Evaluation (LIE), that consists of prompts paired with 4 responses that are {short + (factually) correct, short + incorrect, long + correct, long + incorrect} and train a reward model ($R_{LIE}$) on collected human preferences on examples where the length and correctness are negatively correlated. They then evaluate the reward scores assigned by $R_{LIE}$ (and another reward model trained on a held-out dataset HelpSteer2, $R_{HS2}$), on each of the 4 kinds of responses. They observe that both $R_{LIE}$ and $R_{HS2}$ assign higher scores to the longer responses, regardless of correctness due to the unreliability of the human annotations (both collected judgments for this paper, as well as those in help-steer).

In order to mitigate this issue, the authors propose to augment the traditional learning objective for reward models with a penalty for the annotator's reliability on that particular example. This is incorporated either with the $\beta$ parameter or as a corrective regularization term in the contemporary loss function.

In order to obtain reliability scores for examples, the authors experiment with three different ways: (1) self-reported annotator confidence, (2) a synthetic LLM-assigned grade with CoT prompting, (3) a learned classifier score that involves (a) collecting a dataset of 1000 new preference judgments on examples with a correct answer from various benchmarks - MMLU, BigBench, etc. (b) fitting a transformer-based classifier on a binary label {0/1} based on whether the annotator selected the correct label (c) the probability assigned by this classifier is used as a reliability score for inference.

Empirical results show that incorporating reliability into model training improves reward model predictions in some cases, with uneven trends between the particular ways of assigning reliability scores and incorporation methods.

**Strengths:**

1. The premise for the work, that annotator reliability can be a big factor in reward model performance, is solid and grounded in issues that have also been observed in ML research in other domains. Incorporating signal about this reliability into the model training, essentially connecting the data collection process to the model training, is intuitive.

2. The analysis piece on the LIE dataset is well done and a compelling data contribution given the controlled nature of the collected responses.

3. The proposed mitigation strategies are interesting and pretty novel and can be incorporated into the existing training paradigm of reward models with minimal editing of the objective. Empirically testing each combination of methods of assigning reliability scores and incorporating these during training is helpful to the reader.

**Weaknesses:**

1. The paper structure is hard to follow. The first half is clear with the collection of the LIE dataset as an analysis tool to confirm the proposed issue. The proposed mitigation strategies section could be improved as Section 6.1 is the methods for incorporating the reliability scores, 6.2 is the first 2 ways of obtaining reliability scores, then 6.3 is only the third way i.e. the TRUE dataset and the paper abruptly ends after 6.4 which is the experiments on mitigation without a proper conclusion. This is fixable but the current structure feels incomplete.

2. The aforementioned issue is exacerbated because it's hard to obtain a clear takeaway from the results themselves. For instance, there doesn't seem to be a clear trend between the methods of incorporating the reliability scores via the $\beta$ parameter or the regularization scheme in Table 1 across the three reliability measures. Similarly, there also doesn't seem to be a clear winner among the three schemes of assigning scores, particularly because many of these combinations are outperformed by the normal preference learning and the reliability-aware preference learning with constant scores.

3. The paper lacks many details to reproduce the findings including specifically (a) L.469-473 - the fitting of scores is done to which set of examples? Those from True? (b) L.488-492 - What was the sampling scheme for creating this dataset? How do you verify that each example has a single correct answer? (c) For Table 1, what was the training scheme for the models trained on TRUE? What was the variance across different seeds/hyperparam sweeps?

**Questions:**

1. (This isn't a weakness, just a comment) The name for the TRUE dataset is cute given the other dataset was called LIE, but it feels contrived. 'Testing Reasoning and Understanding Errors' is very generic and reasoning with LLMs is a much more focused research area than sampling examples from various datasets.

2. The proposed mitigation strategies read as 'practical' ways to incorporate the reliability score by minimally modifying the objective. Can you provide some theoretical justification/intuition for the different methods?

3. There's an assumption of generalizability of the reliability scores learned from the TRUE dataset to other domains. This seems reasonable because the scores from TRUE are better than the self-reported confidence and LLM-scores in Table 1 but it is underperformed by the constant score assignments.

---

### Official Review · Reviewer_3Kmu · 2024-11-02

**Soundness:** 2
**Presentation:** 2
**Contribution:** 2
**Rating:** 5
**Confidence:** 4

**Summary:**

This work aims to make preference learning more robust to noise in human preference labeling, specifically focussing on length bias.

As a first step, the authors collect a dataset called LIE (Length Incentivized Evaluations) that contains queries with 4 types of answers: (1) Short Correct, (2) Short Incorrect, (3) Long Correct, and (4) Long Incorrect.
Next, 20 annotators are asked to provide preferences for answer pairs in the dataset, and it is confirmed that annotators prefer longer answers over correct answers (as found in earlier work as well).
Then the authors train a standard reward model on the collected preferences, which has the same length bias.

Finally, the authors try to mitigate the bias by “reliability aware preference learning” (RAPL). The idea is to either adjust the reward or the bolzmann probability based on the difficulty of the questions. The authors experiment with 2 approaches to set the values: (1) Annotator self-reported confidence, and (2) an LLM-based autograder. Calibration is done through another collected dataset (TRUE), that contains annotator answers for questions with known answers.

Based on the experiments, the effectiveness of either method is somewhat unclear. Adjusting the bolzmann probability based on models fine-tuned on the TRUE dataset seems to work best, although the authors mention that “this strategy doesn’t work for just any value as the model trained using the average confidence value doesn’t perform too well” (line 511-512).

**Strengths:**

* **Paper addresses an important problem:** it is important to work with reliable human preference judgements, and thus to quantify the reliability of the judgements, and to mitigate any unreliability when needed.
* **Collected dataset can be useful for future work:** the collected LIE dataset can be used by future work to check how biased annotators or reward models are against length bias.

**Weaknesses:**

* **Effectiveness of the proposed solution is unclear:** many of the proposed solutions seem to rather increase the length bias. For the method that decreases the length bias, the authors write: “this strategy doesn’t work for just any value as the model trained using the average confidence value doesn’t perform too well”. This makes it unclear how well the proposed mitigation strategies work.
* **Especially the second half of the paper is somewhat unclear.** This is enforced as the paper ends somewhat abruptly, without a clear conclusion (the last section is a section called ‘experiments’). I have a few specific questions, that I summarize below in the question box.
* The LIE dataset is useful to quantify length bias, but I would rephrase contribution 2 (“we find that reward models trained on unreliable human feedback tend to place higher weight on obvious proxies like length and less weight on factual correctness”) to “we *confirm* that ...” to do better justice to findings from prior work.

**Questions:**

* It is not entirely clear to me which of the proposed mitigation methods the authors would recommend to use, based on the experimental results?
* The use of the TRUE dataset is somewhat unclear to me, even after reading the appendix. Can the same effect be achieved if a validation set of the LIE dataset had been used to calibrate, given that this dataset also has correctness labels?
* It is hard to connect the plots in Figure 3a, b to the text. Can the authors detail how we can conclude that annotators prefer longer answers, based on the info in these plots?
* About the LIE dataset: in the appendix we can read that one of the authors checked for correctness. How was this done? How many changes were made? How was reliability measured, as one cannot measure inter-annotator agreement with one annotator?
* The abbreviations LC, LI, SC, SI are used without introduction. I assume those stand for ‘Long Correct’, ‘Long Incorrect’, ‘Short Correct’, and ‘Short Incorrect’?

---

### Official Review · Reviewer_2aur · 2024-11-03

**Soundness:** 2
**Presentation:** 3
**Contribution:** 3
**Rating:** 5
**Confidence:** 4

**Summary:**

The work aims to improve the reliability of reward modeling for better model alignment. First, they construct the LIE dataset where incorrect responses tend to be longer. The initial experiment shows that a reward model can be successfully misguided to favor longer incorrect responses. Based on this observation, they propose two methods, Reward Adjustment and Probability Adjustment, to account for the unreliability of preference for reward modeling. Both methods work by deemphasizing unreliable preference pairs in the training loss. The unreliability is estimated by either human or model-generated scores. They also construct the TRUE dataset to further calibrate the reliability measures. Experimental results show that Probability Adjustment with calibrated scores yields the best reward model on both their LIE dataset and in-the-wild dataset HelpSteer2.

**Strengths:**

1. Reward modeling is key to the success of model alignment and this work investigates the reliability of preference labels which is overlooked by previous works.
2. The word design a well-curated dataset LIE and evaluation metric to demonstrate that the unreliable preference labels indeed lead to reward models that favor length over correctness.
3. The work proposes reasonable methods to account for unreliability in reward modeling and verify their effectiveness.

**Weaknesses:**

1. The proposed method seems to heavily rely on human annotation (the TRUE dataset) to calibrate the reliability measure, without which the CLR scores are not higher than the baseline (e.g., the LLM-based methods vs. Normal PL in Table 1).
2. How the reward models would affect the model alignment is not verified. This makes the whole study of this work less motivated.
3. The paper ends a bit rush. I understand there is already extensive discussion throughout the paper but still a better organization of the conclusion session would be helpful.

**Questions:**

1. Could the authors clarify whether they are doing DPO training directly or they only do reward modeling?  If it is the latter case, it seems the method could be applied to DPO as well. I would be curious about the downstream performance then.
2. In Table 2, the CLR score for Normal PL is only 0.02, which is low. Does it mean that the unreliability issue is indeed not severe in realistic datasets?
3. In Eq. (6), what is the purpose of the second term "(1-p) * 0.5"? It seems it is just a constant offset which does not affect the training?

---

### Official Review · Reviewer_soZr · 2024-11-12

**Soundness:** 2
**Presentation:** 2
**Contribution:** 2
**Rating:** 3
**Confidence:** 4

**Summary:**

This paper tackles the problem of learning from unreliable human annotations in RLHF. Specifically, it is assumed that there is some chance that the human label is incorrect / has low confidence. The reliability is then incorporated into the reward learning objective through 1) temperature scaling of the reward and 2) interpolation with a random guessing distribution. To evaluate the proposed approach, a new dataset is also built to measure reliance on length when evaluating answer correctness.

**Strengths:**

- This paper tackles an important problem in the current paradigm of learning from human feedback.
- The LIE dataset is useful for evaluating biases in reward models.

**Weaknesses:**

- There are two proposed ways to incorporate reliability into the objective, however: 1) temperature scaling would not work since it doesn’t change the objective. The model would in fact increase the difference between the rewards to compensate for the temperature. It can be applied at inference time, e.g., when using the RM in RLHF, though. 2) the interpolation method is not novel and has been used in (Chistiano et al, 2017), which is also pointed out by the author.
- The paper seems incomplete, ending abruptly at experiment results. From a presentation perspective, the description of the motivation and intuitive for the proposed methods is quite verbose but the technical details and result description are thin.

**Questions:**

- The TRUE dataset is essentially used to estimate the likelihood of human making an error on a specific question. This seems challenging and depend on annotator skill level and domains. How good is the model at this task and how well does it generalize?
- Given the estimated error probability from TRUE, a simpler baseline would be to just skip examples whose annotation has an error probability beyond certain threshold.

---

### Note · Authors · 2024-11-17

I have read and agree with the venue's withdrawal policy on behalf of myself and my co-authors.